# Assessing a Tourism City from an Ecosystem Services Perspective: The Evaluation of Tourism Service in Liyang, China

Xiangnan Fan and Yuning Cheng *

School of Architecture, Southeast University, Nanjing 210096, China; fan-xiang-nan@seu.edu.cn
* Correspondence: 101004222@seu.edu.cn

**Abstract:** Tourism is an important industry that promotes national economic and social progress. All-for-one tourism is a new concept of regionally coordinated development that uses the tourism industry as an engine to boost resource integration, industrial integration, and social sharing. Tourism service is the main embodiment of cultural ecosystem services for all-for-one tourism cities. Taking the city of Liyang in China as an example, this paper used a combination of GIS spatial analysis and big data text mining to evaluate tourism service from three aspects: the quality of tourism resources, the comprehensiveness of tourism service facilities, and the satisfaction of tourists. The results show that (1) tourism service is better in the northwestern and eastern areas of the city, while it is lower in the northeastern and southwestern parts; (2) the hotspot areas should focus on improving tour routes, transport capacity, and excessive charges; the cold spot areas should work on ecological restoration and creating new tourism attractions by combining the local industries; and (3) rural tourism integrating agriculture and visitation should be highlighted as a key growth point to improve the city's tourism service function.

**Keywords:** tourism service evaluation; ecosystem services (ESs); AHP-entropy weight method; GIS spatial analysis; text mining





## 1. Introduction

Ecosystems are the foundation for human survival and development, providing the environmental conditions and material support necessary for human existence. The benefits that humans derive from the natural environment are defined as ecosystem services (ESs) [1]. The basic ideas behind ESs germinated in the late 1960s. The term "ecosystem services" was coined in 1982 by ecological scientist Walter Westman, bringing the concept into wider use [2]. In 1997, ecological economists Robert Costanza et al. published a landmark paper describing a methodology for valuing ecosystem service functions and estimating the annual economic value of the world's ecosystem services to be $33 trillion [3]. This research has led to the recognition that public environmental resources, such as clean water and biological resources, are limited and valuable, and has made people aware of the importance of the public goods provided by the ecological environment [4]. The Millennium Ecosystem Assessment, conducted in 2005, systematically studied the relationship between ESs and human well-being and focused on the assessment of ecosystem service functions. The project classified ESs into four categories: provisioning, regulating, cultural, and supporting services, and this classification system has since become widely used in environmental science, economics, and policy.

Cultural ecosystem service (CES) refers to the non-material well-being that humans obtain from ecosystems, including aesthetic inspiration, cultural identity, sense of home, and spiritual experiences [1]. Cultural ecosystem services (CESs) are directly experienced and subjectively valued by people who benefit from them [5]. They are more closely related to human well-being [6,7] and ecosystem sustainability [8]. However, it is difficult to identify

and evaluate CESs, since the value of these intangible benefits depends not only on the physical landscape and infrastructure but also on public demand and social culture [9]. Because of this challenge, the evaluation of CESs has become a hot topic in recent years. The evaluation methods are generally divided into value accounting [10–13], the land use matrix method [14–16], the participatory method [17,18], and model simulation [19–21]. Currently, most evaluations of CESs focus on assessing the overall CESs from the perspective of service supply and demand or synthesizing the overall results through simple evaluations of each service category.

Some studies focus on the evaluation of a particular type of CES, mostly recreation service [22–24] and aesthetic service [25–27], which are easier to quantify. Very little of the literature deals with tourism service, even though tourism is a significant contributor to ecosystem services. A very important reason is that there is some controversy about whether the tourism service belongs to cultural services. Some authoritative institutional reports that serve as the basis of research provide relevant elaborations on tourism. The Millennium Ecosystem Assessment (MEA) report uses the term "recreation and ecotourism" as an example to illustrate the classification of cultural services [1] (p. 7). The Economics of Ecosystems and Biodiversity (TEEB) report clearly lists "tourism" as a category under cultural services when describing the classification of ESs [28] (p. 4). Although the two reports classify tourism as a cultural service, the term "ecotourism" is predominantly used throughout the text, suggesting that only tourism closely related to the natural environment is recognized. Moreover, many statements consider tourism as an industry that depends on ESs or describes it as a beneficiary of ESs. As Josep Pueyo-Ros has stated in his paper, tourism is treated in a "schizophrenic" manner [29], being considered on the one hand as a cultural ecosystem service, and on the other hand as a "nature-based consumptive industry". In the SEEA (System of Environmental Economic Accounting) report, the services that benefit visitors/tourists are classified as recreation-related services, and tourism is not directly mentioned [30]. To support the SEEA revision, the European Environment Agency (EEA) has published the Common International Classification of Ecosystem Services (CICES). The classification of ESs in this report is more systematic and complex. Although tourism is not directly mentioned in the category of cultural services, some of the elaborations in the report are instructive as to whether tourism can be considered a cultural service. The report regards cultural services as "the environmental settings, locations or situations" that can involve "individual species, habitats, and whole ecosystems", which is a more specific and detailed definition [31] (p. 10). According to this definition, especially for tourism cities, visitations are the most prominent and dominant setting of the city. In other words, tourism can exist as a cultural service for tourism-oriented cities. It also states that "The settings can be semi-natural as well as natural settings (i.e., can include cultural landscapes) providing they are dependent on in-situ living processes" [31] (p. 10). This indicates that the report no longer emphasizes only completely natural environments, but that semi-natural environments, managed ecosystems, and even cultural landscapes can provide cultural services. Therefore, tourism does not need to be limited to whether it is nature-based or not. For cities like Liyang that are striving to develop all-for-one tourism, all kinds of tourism attractions should be included in the evaluation of tourism service.

At present, there is not a complete theoretical framework or evaluation system for tourism service. Evaluations related to tourism service are often from the perspective of the tourism industry, focusing on tourism performance [32–34], tourist behavior preference [35–37], tourism resource potential [38–40], tourism service quality [41–43], and tourism service economic valuation [44–46]. The evaluation methods mainly include monetary approaches [10,44], questionnaire and interview surveys [36], evaluation by constructing an indicator system [32,33,47], model simulation [48,49], and social media data mining [50,51]. Although monetary quantification is popular and convenient, it is always subjective and difficult to capture non-use values, such as existence or bequest values, which are often important components of the total economic value [52]. Questionnaire

and interview surveys are more conventional and are prone to random errors with small sample data. The model simulation approach draws mainly on the Recreation module of the InVEST model and the SolVES model. However, the InVEST model often cannot accurately reflect the actual situation of the study area. This is because the geotagged photos used in the "Recreation and Tourism" tool come only from the Flickr website, which is not universally used in all countries around the world, and the database only covers the period from 2005 to 2017. The SolVES model can create a statistical model between social survey data and natural environmental variables, but the model's single environmental parameter setting will cause different landscape types to use the same environmental variables, which ultimately leads to inaccurate results [53]

As tourism is an important driving force for integrated urban–rural development and an important platform for external communication, China has vigorously developed the tourism industry since the reform and opening up in the 1980s, becoming the largest domestic tourism market and one of the world's leading tourism economies. In 2015, the National Tourism Administration first proposed the concept of "all-for-one tourism", which considers the whole area as a tourism area and drives the area's development with well-known attractions and rural tourism hotspots, activating the regional recreation potential [54]. Since then, local governments across the country have been exploring ways to integrate tourism with various industries and utilize existing tourism resources to promote regional development [54,55]. For cities where tourism development is very dependent on natural resources, the priority is to ensure a good ecological environment. Therefore, it is necessary to evaluate ESs by taking tourism service as the dominant type of cultural service. Evaluating tourism service from the perspective of ESs is useful for prioritizing the maintenance of an ecosystem's safety and stability, and also helps propose the city's development strategies for the organic integration of the ecosystem and tourism industry.

In view of the great significance of assessing tourism cities from an ecosystem services perspective and the lack of tourism service studies, this paper tried to construct a tourism service evaluation system by taking Liyang as an example. Located in Jiangsu province, Liyang is a typical all-for-one-tourism city, with 25% of the area covered by ecological space, 60% by agricultural space, and 15% by urban and built-up space. Large-scale scenic areas are mainly located in the city's ecological space, while tourism spots are scattered in the agricultural space and concentrated in built-up areas. Rural tourism is a new development mode to integrate urbanization and all-for-one tourism. Since 2013, Liyang has been committed to creating a rural development pattern of "ecological home and leisure land" to comprehensively promote rural tourism. By evaluating the city's tourism service, current deficiencies and strengths could clearly be seen and corresponding improvement suggestions were proposed. The specific objectives of the research include (1) categorizing tourism service as an ecosystem service; (2) enriching the evaluation of cultural ecosystem services; and (3) providing planning guidance for Liyang's future development from the perspective of tourism industry and ecosystem integration.

## 2. Materials and Methods

### 2.1. Study Area

Liyang in Changzhou City is a county-level city located at the intersection of Jiangsu, Zhejiang, and Anhui provinces in the Yangtze River Delta region (Figure 1). Covering an area of 1535 km$^2$, Liyang is rich in landscape resources and has a varied topography, including low mountains, hills, polders, plains, rivers, and lakes. In addition to its natural resources, the city also has rich historical and cultural resources with a long history of more than 2000 years.

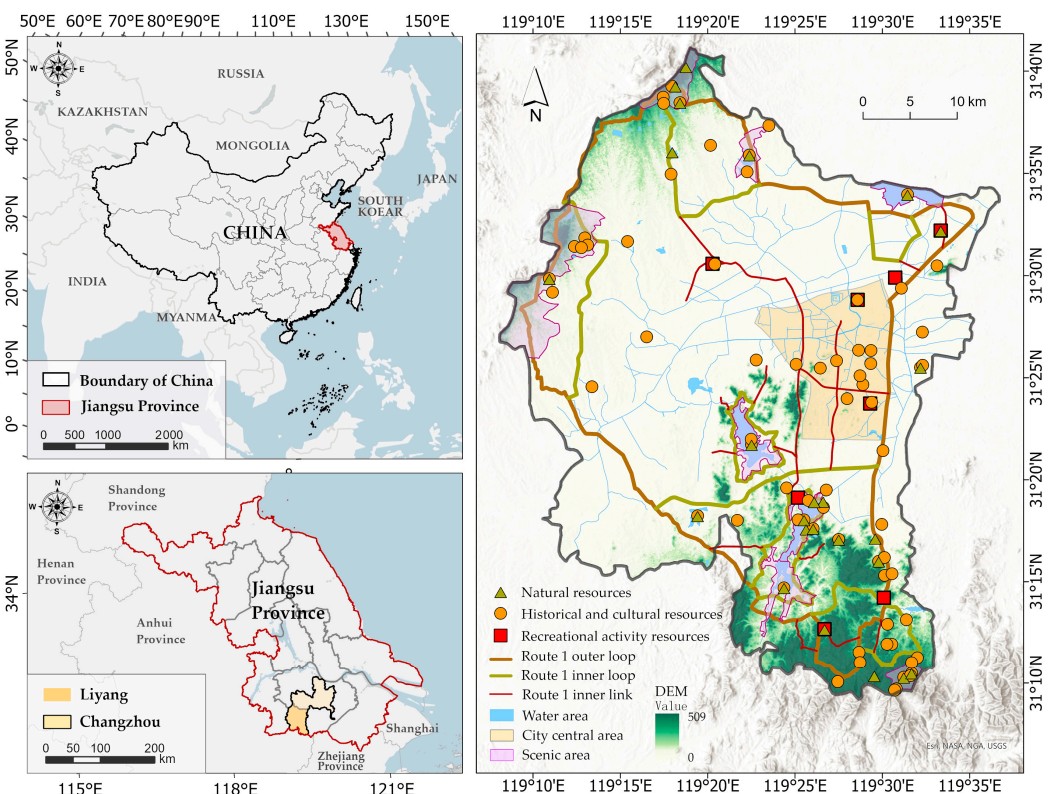

**Figure 1.** Location of the study area and distribution of tourism resources.

As early as the 1990s, Liyang has begun to take tourism as a major industry. Until now, with all-for-one tourism as the driving force and ecological innovation as the focus, urban and rural areas in Liyang are coordinated to promote the whole region's integrated development. In 2020, Liyang was successfully selected as the second batch of National All-for-one Tourism Demonstration Areas. The overall pattern of "three mountains and two lakes surrounding the central city" was formed, consisting of Wawushan Scenic Area, Caoshan Tourist Resort, and Changdang Lake in the north, and Nanshan Bamboo Sea and Tianmu Lake in the south [56]. The city's main tourism attractions are diverse and distinctive, with a general distribution of dispersal throughout the area and concentration in the central city, as shown in Figure 1. According to the national standard [57] and previous studies [40,58], these attractions are divided into three categories: natural resources, historical and cultural resources, and recreational resources. Liyang Route1 is a provincial scenic byway with a total length of 365 km, connecting various scenic spots and attractions, and has initially formed the "grand tourism" ambition of "Variable views everywhere, unique landscape every path".

### 2.2. Data Sources and Processing

The basic data for this research include location data and big data on tourism reviews (Table 1). The location data obtained from Baidu Maps 18.0.0 (accessed on 26 July 2023) and OpenStreetMap 18.0.0 (accessed on 10 May 2023) were georeferenced as the Albers Conical Equal Area projection in ArcGIS 10.8.2. The big data for tourism reviews were collected using the Octopus web crawler, with a cut-off date of 20 July 2023. After classification, cleansing, and analysis, the processed data were geocoded to the corresponding spatial locations of tourism attractions using ArcGIS.

**Table 1.** Basic data.

| Category | Data | Data Source |
|---|---|---|
| Location data | Tourism resources<br>POIs of tourism service facilities | Baidu Maps |
| | Road network | OpenStreetMap |
| Big data for tourism reviews | Number of check-in photos at attractions | Ctrip (www.ctrip.com (accessed on 18 July 2023)), Mafengwo (www.mafengwo.cn (accessed on 18 July 2023)), Tongcheng Travel (www.ly.com (accessed on 18 July 2023)) |
| | Tourism review texts<br>Popularity ratings of attractions | |

*2.3. Methods*

2.3.1. Determination of the Evaluation Unit

The landscape pattern has obvious scale dependence. Granularity is the smallest pixel constituting the landscape. Too small a granularity will easily produce local information redundancy and fail to reflect the overall characteristics of the landscape, while too large a granularity will produce information loss. Therefore, it is necessary to first determine the optimal granularity and use it as the evaluation unit. The evaluation of area information conservation uses the principle of minimum loss of landscape area in the scale transformation process to identify the optimal granularity domain. The improved area-change evaluation index model proposed by Xu et al. [59] was used to calculate the area loss index. The formulas are as follows:

$$L_i = (A_i - A_{bi}) / A_{bi} \times 100 \tag{1}$$

$$S_i = \sqrt{\frac{\sum_{i=1}^{n} L_i^2}{n}} \tag{2}$$

where $L_i$ is the relative value of area loss, $A_i$ and $A_{bi}$ are the area of the *i*-th landscape before and after scale transformation, $S_i$ is the area loss index for the study area, and *n* is the number of landscape types. The larger $S_i$, the greater the area change, and the worse the accuracy of the area after scale transformation.

ArcGIS was used to generate a series of raster images with different resolutions from 50 m to 1200 m at 50 m intervals based on the land use of the study area. Taking the granularities as the horizontal coordinate and the corresponding area loss indexes as the vertical coordinate, the relationship curve was plotted. From Figure 2, it can be seen that between granularities of 50 m and 250 m, the area loss index remains stable below 0.005. Between 250 to 800 m, it fluctuates slightly between 0.005–0.05. After 800 m, it fluctuates dramatically. Overall, the area loss index is at its lowest level (less than 0.02) at granularities of 50–250 m, 400 m, and 600 m, which are the optimal granularity domains.

The analysis of the granularity effect of landscape pattern indexes is to determine the optimal granularity by identifying the inflection points of index curves. By calculating the landscape pattern indexes at different granularities and based on the coefficient variation, six indexes that are sensitive to granularity changes were selected: number of patches (NP), patch density (PD), connectance index (CONNECT), proportion of like adjacencies (PLADJ), edge density (ED), and landscape-shape index (LSI). The values of landscape pattern indexes under different granularities were calculated using Fragstats 4.2 and the granularity effect curves were plotted. As shown in Figure 3, NP, PD, PLADJ, ED, and LSI decrease with increasing granularities, showing a trend of rapid decline from 50 m to 400 m, followed by a slow decline. The CONNECT value decreases rapidly from 50 m to 150 m, remains stable between 150 m and 400 m, and then fluctuates drastically up to 600 m, after which it remains at 0.

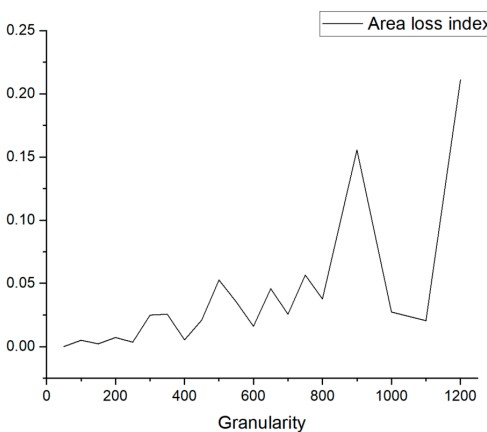

**Figure 2.** Landscape area loss index at different granularities.

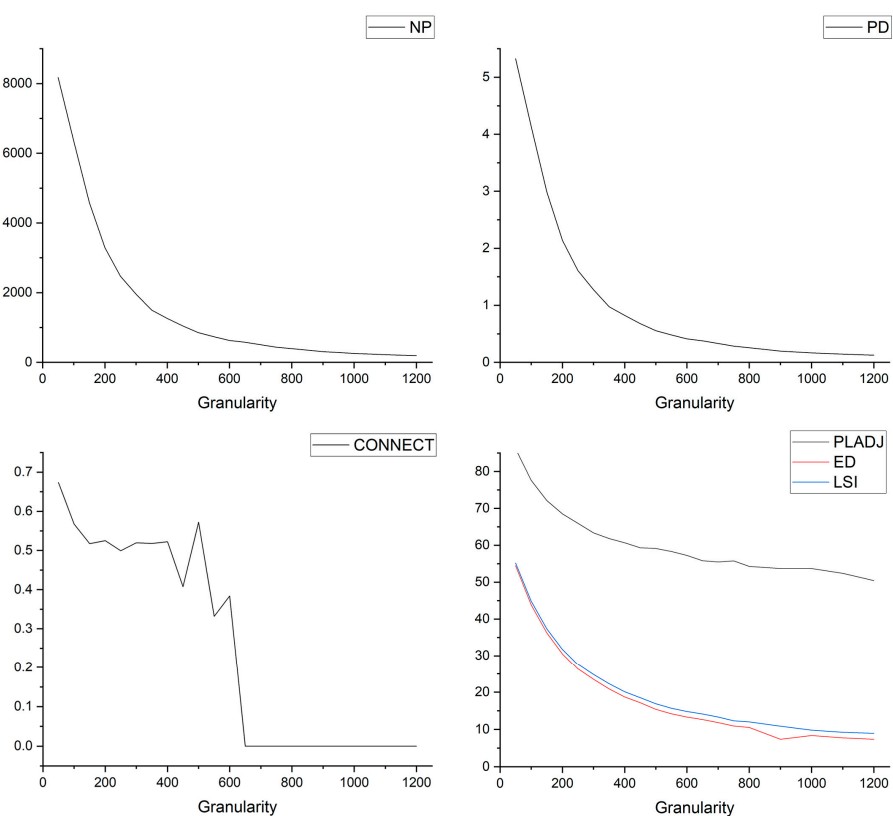

**Figure 3.** Landscape indexes at different granularities.

Considering both the area loss and landscape pattern index granularity effect, 400 m is the optimal granularity that can better reflect the spatial information of the landscape pattern. Therefore, the grid cell of 400 m × 400 m was taken as the evaluation unit, and a grid network was created using the fishnet tool in ArcGIS, with a total of 9949 units overlapping with the study area (Figure 4). On this basis, the spatial visualization of the evaluation results was achieved by calculating the tourism service of each unit.

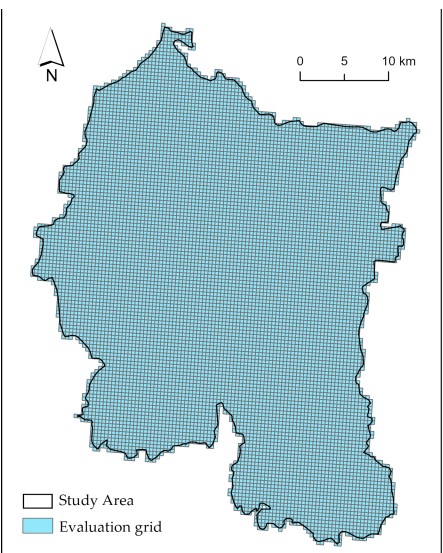

**Figure 4.** Evaluation grid network.

2.3.2. Construction of the Evaluation Indicator System for Tourism Service

1.　　Selection of evaluation indicators

　　The evaluation indicator system for tourism service provides a comprehensive framework for assessing service supply and tourist demand. Tourism service supply is reflected in the condition of tourism resources and tourism service facilities, while tourist demand is reflected in visitors' sentiment tendencies and satisfaction levels. According to data availability and reference to previous research [39,40,60], seven factors were selected to construct the evaluation system (Table 2). The quality of tourism resources refers to the attributes and attractiveness that make the tourism sites appealing and worth visiting for tourists, including the density, fame, and popularity of the tourism resources. The comprehensiveness of tourism service facilities reflects the completeness of amenities and infrastructure to serve tourists at a destination, mainly consisting of tourism service POI density and transport accessibility. Tourist satisfaction is an index of the extent to which tourists' expectations and needs are met by their visitation to a destination, as measured by the number of check-in photos at attractions and the sentiment orientation of tourist reviews.

**Table 2.** Evaluation indicator system for tourism service.

| Target | Indexes | Weight | Factors | Weight |
|---|---|---|---|---|
| Tourism service | Quality of tourism resources | 0.2427 | Density | 0.0953 |
| | | | Fame | 0.0749 |
| | | | Popularity | 0.0725 |
| | Comprehensiveness of tourism service facilities | 0.581 | Tourism service POI density | 0.063 |
| | | | Transport accessibility | 0.518 |
| | Tourist satisfaction | 0.1763 | Number of check-in photos at attractions | 0.085 |
| | | | Sentiment orientation of tourist reviews | 0.0913 |

2.　　Calculation of Indicator Weights

　　The weights of indicators were determined using the AHP-Entropy method, which is a combination of the Analytic Hierarchy Process (AHP) and entropy weighting methods. Since it combines subjective judgment and objective data dispersal, the evaluation result is more realistic and reliable [61]. The AHP weights were calculated using yaahp10.1 in the following steps [61,62]: (1) define the goal and organize the criteria into a hierarchy with the goal at the top; (2) conduct pairwise comparisons to assign numerical values to the relative importance of criteria and alternatives in a decision problem, using a 1–9 scale

and a comparison matrix; and (3) calculate the consistency ratio to check whether the judgments are consistent enough to proceed; the tested consistency ratio is 0, which is less than 0.1, indicating that the results of the calculation of the model's indicator weights are reasonable; and (4) calculate the weights of each criterion based on the results of the pairwise comparisons. The final weights of the seven individual factors were obtained as $w'_{ij}$, where $i$ is the $i$-th factor and $j$ is the $j$-th grid.

The entropy weighting method is a way to calculate the weights of the factors based on the information theory. Entropy is a measure of the degree of disorder in a system and can be used to assess the degree of dispersion of an indicator. The lower the entropy of an indicator, the more information it provides, and the higher its weight should be. First, it is necessary to standardize each factor to obtain a dimensionless value [63,64]. The standardization formulas for positive and negative factors are as follows:

$$Y_{ij} = \frac{X_{ij} - X_{imin}}{X_{imax} - X_{imin}} \tag{3}$$

$$Y_{ij} = \frac{X_{imax} - X_{ij}}{X_{imax} - X_{imin}} \tag{4}$$

where $Y_{ij}$ is the standardized value of factor $i$ in grid $j$, $X_{ij}$ is the original value of factor $i$ in grid $j$, and $X_{imax}$ and $X_{imin}$ are the maximum and minimum values of factor $i$ in grid $j$. The weight calculation formula is as follows:

$$w_{ij} = \propto w'_{ij} + (1 - \propto)w''_{ij} \tag{5}$$

where $\propto$ is the linear combination coefficient and is equal to 0.5, and $w'_{ij}$ and $w''_{ij}$ are the weights obtained from the APH and entropy weighting methods, respectively.

### 2.3.3. Text Mining

Since tourist reviews, user ratings, and other texts on travel websites can directly reflect tourists' experiential perceptions, the text mining method was used to collect and analyze tourism big data, providing data sources for evaluating tourist satisfaction. The Octopus web crawler was used to collect travel review data from Ctrip (www.ctrip.com (accessed on 18 July 2023)), Mafengwo (www.mafengwo.cn (accessed on 18 July 2023)), and Tongcheng Travel (www.ly.com (accessed on 18 July 2023)), including the information on review title, review content, ratings, time, and photos. The data collection period was until July 2023. After removing duplicates and filtering out invalid reviews, a total of 51,916 valid reviews were obtained.

ROST Content Ming 6.0 is a text mining software developed by the Virtual Learning Team from Wuhan University, which can help with word segmentation, word frequency statistics, sentiment analysis, and social network visualization. First, the reviews of each attraction were segmented into words and word frequencies were counted. Then, the adjectives, adverbs, verbs, and nouns that could reflect the tourists' attitudes were filtered out and added to the HOWNET dictionary. On this basis, the sentiment analysis was carried out. In addition, high-frequency positive and negative words were visualized as word clouds to intuitively reflect tourists' attitudes.

### 2.3.4. Kernel Density Analysis

Kernel density analysis is a statistical method for estimating the probability density function of a given set of data points. It was used in this study to estimate the distribution of tourism attractions and tourism service infrastructures. The calculation formula is as follows:

$$f(x) = \frac{1}{nh} \sum_{i=1}^{n} K\left(\frac{x - x_i}{h}\right) \tag{6}$$

where $f(x)$ is the estimated density at point $x$, $n$ is the number of points, $h$ is the bandwidth and is set as the default value by the algorithm in ArcGIS [65], $x_i$ is the $i$-th data point, and $K$ is the kernel function.

### 2.3.5. Accessibility Measurement

The Accumulative Cost Distance model was used to measure accessibility by calculating the minimum cumulative cost from a cell to the source locations via a cost surface. The cost surface was a 400 m grid map with a rasterized road network. According to the Technical Standard of Highway Engineering of the People's Republic of China (JTGB01-2014), the time cost values of the grids were calculated with the speed of different road levels (Table 3).

**Table 3.** Speed and time cost values for different road levels.

| Road Levels | Motorway | Level 1 | Level 2 | Level 3 | Level 4 | Pedestrian | Other |
|---|---|---|---|---|---|---|---|
| Speed (km/h) | 100 | 80 | 60 | 40 | 20 | 10 | 5 |
| Time cost value (min) | 0.6 | 0.75 | 1 | 1.5 | 3 | 6 | 12 |

### 2.3.6. Hot Spot Analysis

Hot spot analysis is a spatial analysis and mapping technique that identifies clusters with high values (hot spots) and low values (cold spots). The Getis-Ord Gi* statistic in ArcGIS was used to identify the hot spots and cold spots of tourism service. The calculation formula is as follows:

$$G_i^*(d) = \frac{\sum_{i=1}^{n} W_{ij}(d) x_i}{\sum_{i=1}^{n} x_i} \tag{7}$$

where $n$ is the total number of grids; $x_i$ is the attribute value for grid $x$; and $W_{ij}(d)$ is the spatial weight between grid $i$ and grid $j$ based on distance $d$.

## 3. Results

### 3.1. Evaluation of the Quality of Tourism Resources

High-quality destinations tend to attract more visitors, generate more revenue, and increase visitor satisfaction and local pride. Figure 5 shows the assessment of the quality of tourism resources in the study area from three aspects. The density analysis shows the spatial distribution and the aggregation degree of tourism resources. By concentrating and combining different types of tourism attractions, destinations can offer a wide range of experiences and activities. Therefore, the denser the tourism attractions, the stronger the aggregation effect and the higher the quality of the destinations. Government departments at all levels, such as the Ministry of Culture and Tourism, Ministry of Agriculture and Rural Affairs, Provincial Housing and Urban–Rural Development, etc., have evaluated and leveled tourism attractions according to some specific criteria. The reputation indicator was scored according to the level of the attractions, with the following principles: national-level attractions scored 8–10, provincial-level attractions scored 7–8, city-level attractions scored 5–7, Internet hotspot attractions scored 2–5, and general attractions scored 1–2. The popularity scores of the tourism resources were obtained from the tourism websites, which were comprehensively calculated by the number of user views and reviews.

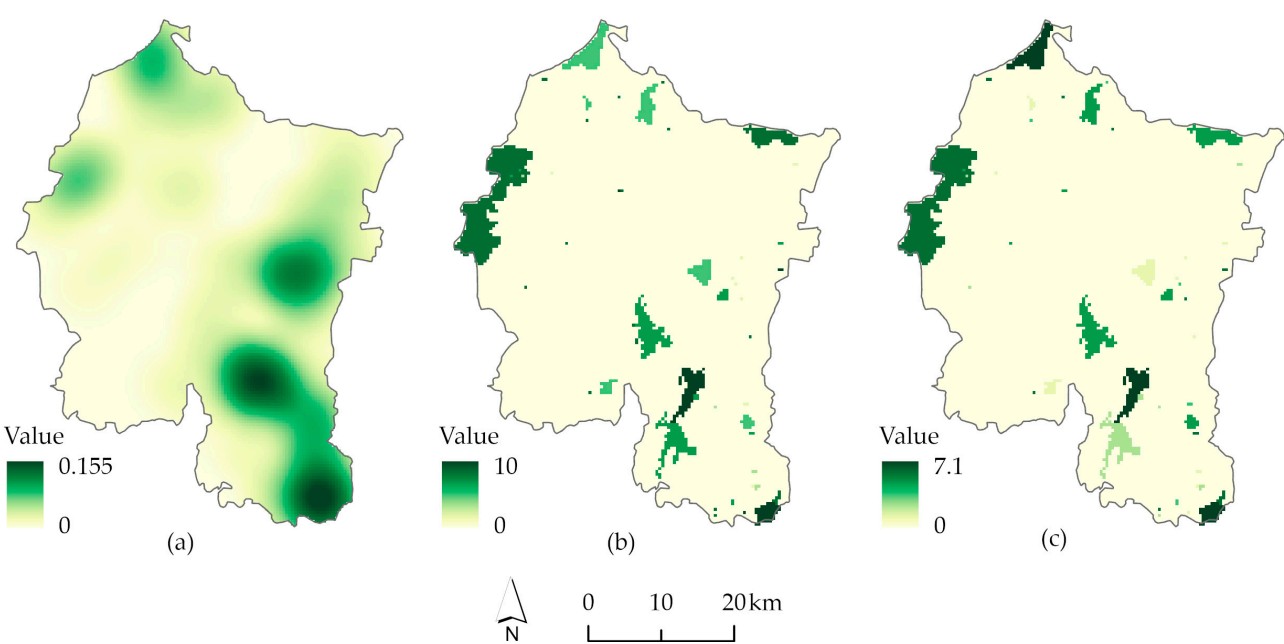

**Figure 5.** Quality of Tourism Resource Evaluation. (**a**) Density; (**b**) reputation; (**c**) popularity.

### 3.2. Evaluation of the Comprehensiveness of Tourism Service Facilities

Tourism service facilities are important in providing a positive experience for visitors and supporting the tourism industry, covering the aspects of accommodation, medical care, catering, recreation, and transport. The distribution of tourism service POIs and the road network is shown in Figure 6. Accommodation facilities are hotels and other types of lodging facilities that provide overnight accommodation for visitors. Medical facilities are hospitals, clinics, and pharmacies that provide medical treatment services to visitors. Catering facilities are restaurants, cafes, bars, and other establishments that provide food and drink services. These tourism POIs are used for density analysis and the road network is used for transportation accessibility evaluation (Figure 7).

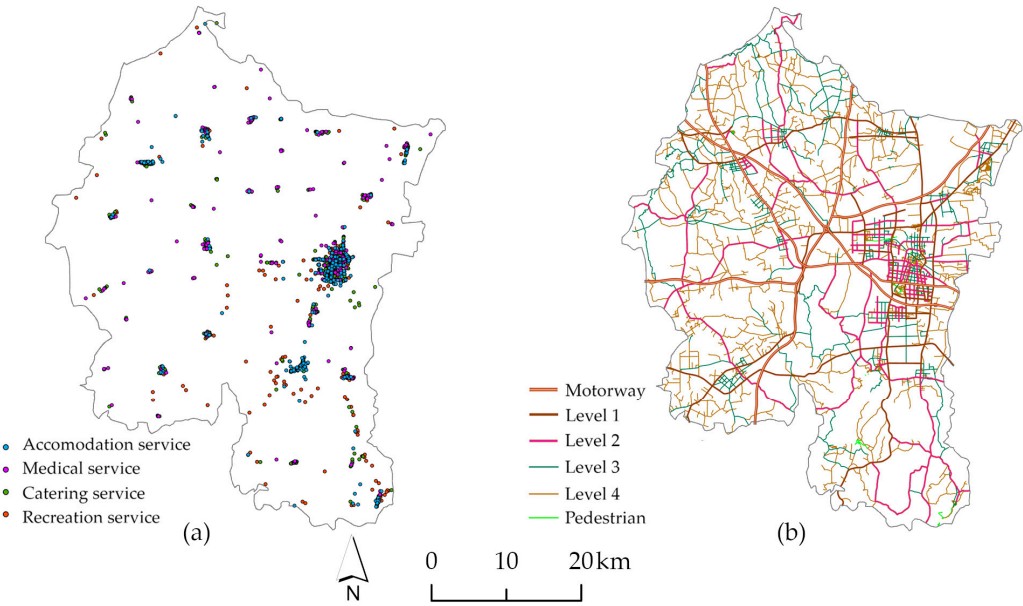

**Figure 6.** Distribution of tourism service facilities. (**a**) Tourism service POIs; (**b**) road network.

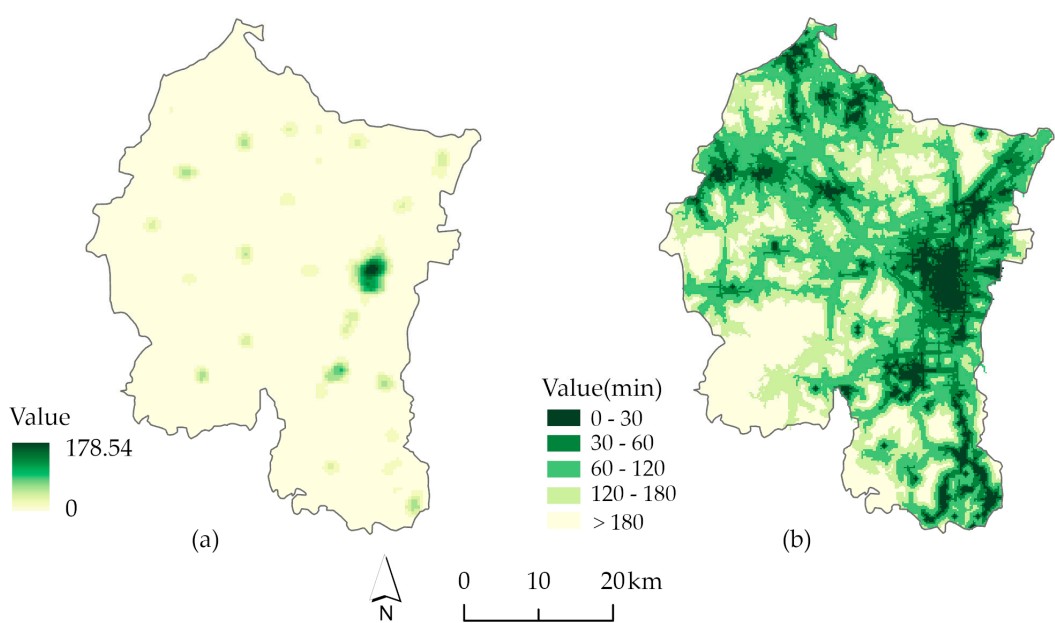

**Figure 7.** Evaluation of the comprehensiveness of tourism service facilities. (**a**) Tourism service POI density; (**b**) transport accessibility.

### 3.3. Evaluation of Tourist Satisfaction

The evaluation of tourist satisfaction was based on the reviews crawled from Ctrip, Mafengwo, and Tongcheng Travel. The more willing the tourists were to visit, the more photos they would take, and the more comments they would make. Figure 8a shows the total number of check-in photos from the three online travel websites for each tourism attraction. Using ROST CM6.0 to analyze the sentiment orientation of tourist comments, the number and proportion of positive, neutral, and negative comments for each attraction were obtained. To make the results of the sentiment analysis more intuitive in spatial representation, the positive, neutral, and negative sentiment orientations were first assigned scores of 2, 1, and $-2$, respectively. Then, the proportion of each sentiment type at the specific attraction was multiplied by its corresponding score and summed up, resulting in the sentiment orientation score for each attraction, as shown in Figure 8b. The higher the score, the more positive the reviews.

### 3.4. Comprehensive Evaluation of Tourism Service

As the units of the indicators are different, the evaluation result of each indicator was first normalized. Then, the normalized indicators were comprehensively superimposed according to the weight values obtained using the AHP-entropy method. Finally, the overall evaluation was divided into five levels based on the natural breakpoint method, as shown in Figure 9. The high-value areas (areas at high and higher levels) are distributed in the northwestern edge and the east of the city, while the low-value areas (areas at low and lower levels) are mainly distributed in the northeastern and southwestern parts of the city. According to the statistics of the area of each level, about 1/4 of the city area is at the low and lower levels, over 27% is at the high and higher levels, and 47.08% of the area is at the medium level.

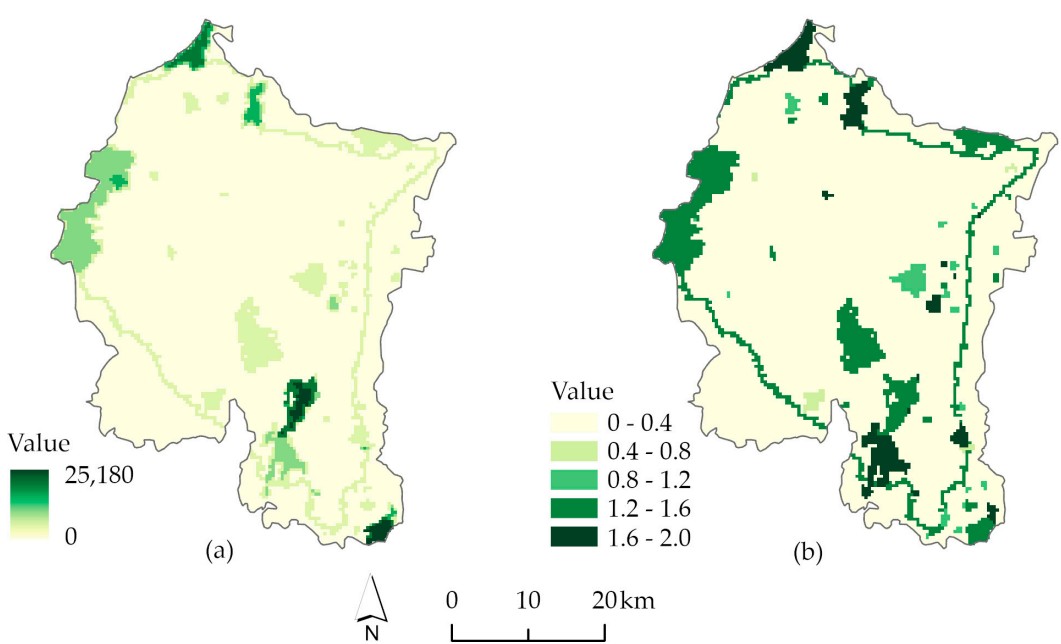

**Figure 8.** Evaluation of tourist satisfaction. (**a**) Number of check-in photos at attractions; (**b**) sentiment orientation of tourist reviews.

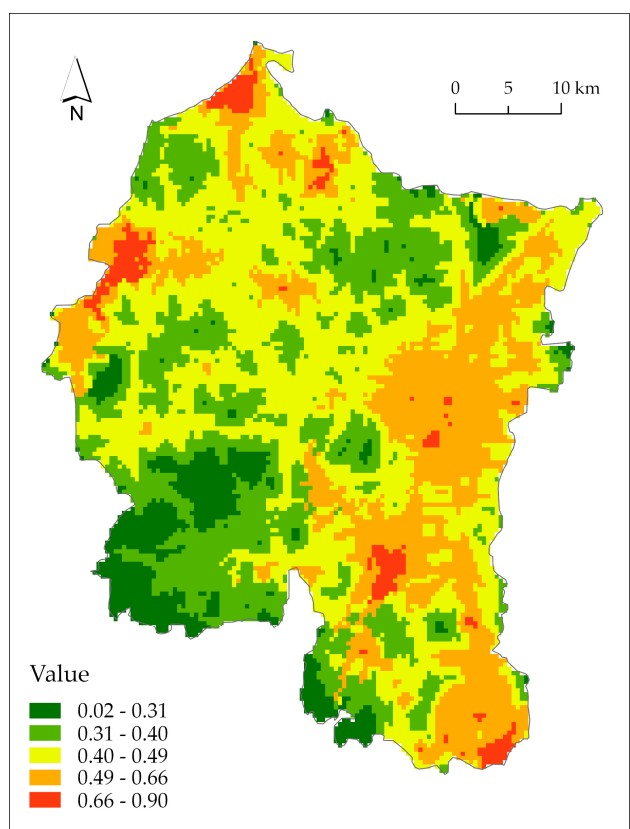

**Figure 9.** Comprehensive evaluation of tourism service.

From the area proportion of each administrative division (Table 4), it can be seen that Licheng Subdistrict, located in the main urban area, has the highest tourism service function, with more than 91% of its area at the high and higher levels. Guxian Subdistrict, also located in the main urban area, has the second highest tourism service, with a high-value area of more than 53% and a low-value area of less than 3%. This is due to the concentration

of resources and convenient transport in the central urban area. Daibu Town and Daitou Town also have a relatively high tourism service, with a high-value area of about 57% and 48%, respectively. Shezhu Town has the lowest tourism service, with only about 2% of its area at the high and higher levels and over 78% of its area at the low and lower levels, mainly due to its lack of tourism resources. Bieqiao Town is also in the same situation with a small proportion (11.66%) of high-value area and a large proportion (33.34%) of low-value area. The rest of the towns have a medium level of tourism service.

**Table 4.** Statistics of the area proportion of tourism service by different levels.

| Area Proportion (%) | | Lower Level | Low Level | Medium Level | High Level | Higher Level |
|---|---|---|---|---|---|---|
| | Whole city | 4.07 | 21.28 | 47.08 | 24.72 | 2.85 |
| | Zhuze Town | 0.00 | 4.59 | 67.34 | 22.39 | 5.68 |
| | Shangxing Town | 1.80 | 21.69 | 53.05 | 17.70 | 5.76 |
| | Nandu Town | 0.65 | 19.46 | 77.20 | 2.68 | 0.00 |
| | Shezhu Town | 20.24 | 58.33 | 19.49 | 1.94 | 0.00 |
| | Bieqiao Town | 0.18 | 33.15 | 55.00 | 9.42 | 2.24 |
| City divisions | Shanghuang Town | 4.07 | 12.44 | 49.79 | 33.48 | 0.21 |
| | Daitou Town | 0.91 | 11.74 | 39.25 | 48.09 | 0.00 |
| | Kunlun Subdistrict | 0.00 | 11.64 | 58.98 | 29.38 | 0.00 |
| | Licheng Subdistrict | 0.00 | 0.10 | 8.15 | 88.18 | 3.57 |
| | Guxian Subdistrict | 0.00 | 2.84 | 43.69 | 52.06 | 1.42 |
| | Tianmu Lake Town | 5.37 | 20.85 | 42.56 | 27.13 | 4.10 |
| | Daibu Town | 0.07 | 6.61 | 36.24 | 52.42 | 4.66 |

To better understand the spatial clustering, the local spatial autocorrelation analysis was used to identify hot and cold spots. As shown in Figure 10, the hot spots are mainly concentrated in Wawushan Scenic Area and Bieqiao Village in the northern part of the city, Caoshan Resort in the west, Tianmu Lake and Nanshan Bamboo Sea in the south, and the central area of the city. The cold spots are mainly concentrated in the southwest.

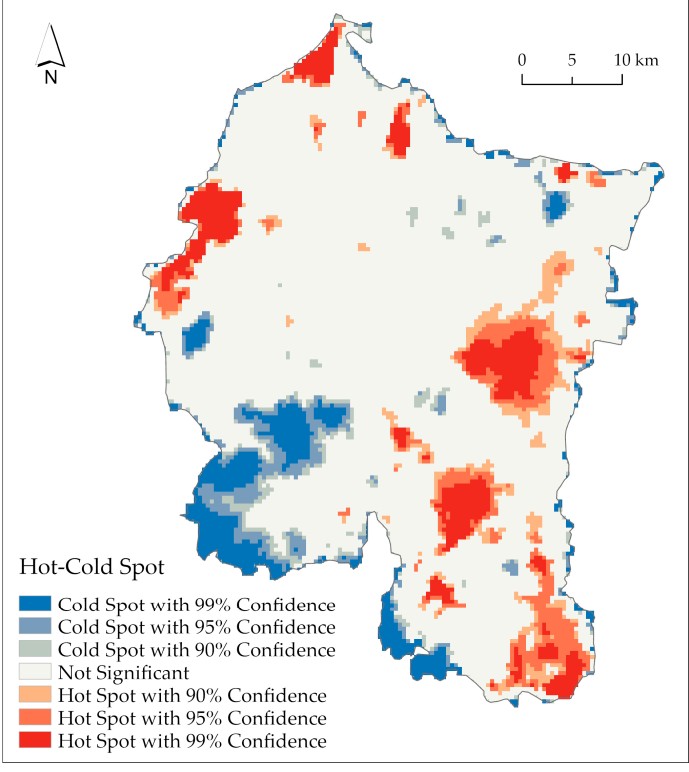

**Figure 10.** Hot–cold spot analysis.

Word cloud graphs can intuitively and clearly reflect tourists' perceptions of scenic spots. The higher the frequency of a word, the larger its font size and the closer it is to the center of the image. First, the positive and negative words corresponding to each hotspot area were summarized, then synonyms were merged and word frequencies were counted, and finally the words with the highest frequencies were extracted to generate word cloud graphs. As shown in Figure 11, the red word cloud graphs were generated from positive words, and the blue ones from negative words. Overall, the high-frequency positive words in each hotspot area are quite similar, mainly reflecting that the attractions in the area have beautiful scenery and fresh air, which are pleasing and worth visiting. The high-frequency negative words have different evaluation tendencies and reflect where the scenic areas need to be improved in the future. Wawushan Scenic Area has few negative words, mainly reflecting the fact that the mountain is too short and small to be particularly attractive. Currently, people mainly pass through to see the mountain forest scenery, and there is little potential for staying. In the future, it may be possible to expand the range of mountain resources and develop rural tourism by combining the surrounding farmlands and villages, in order to enrich the tourism offerings of this scenic area. Caoshan Resort, called Colorful Caoshan, is a provincial-level tourism resort that integrates ecological agriculture, rural culture, sports and leisure, health and wellness, and conference and exhibition. The high frequency of negative words reflects tourists' dissatisfaction with the theme parks. The two amusement parks, one themed on cars and the other on trains, require not only an admission fee but also additional payment for each play item, which is expensive and makes tourists feel that they have been fooled. Tourism attractions in the central city are numerous but small, mainly consisting of urban parks and squares. The target audience is mainly local residents, with only a few visitors from other places. There is a high demand for the usability of these public spaces, and negative experiences of these attractions tend to focus on withered trees and dilapidated facilities. Therefore, it is necessary to strengthen the management of greenery and the maintenance of service facilities in the central city. Negative reviews of Tianmu Lake focus on high fees and unreasonable tour routes. Many comments mention that the required ticket and boat ride alone cost up to 180 yuan, and other services such as parking, food, and storage are also expensive. The tour route is designed so that the islands around the lake can only be reached by boat. It takes a long time to queue for the boat, and the ticket only covers a one-way trip. The return trip involves a long walk up the mountain to get back to the entrance, leaving visitors exhausted. The negative comments about the Nanshan Bamboo Sea reflect problems such as high fees, insufficient transport capacity, and over-commercialization. In addition to the high entrance fee, visitors who travel by car also have to pay a parking fee of 10 yuan per hour. Some major attractions, such as the Panda Pavilion and the Jiming Village, can only be reached by transport within the scenic area, which requires a significant additional fee. Due to the limited transport capacity and large crowds, visitors often have to wait in long queues for up to two hours just to take a two-minute cable car ride. In addition, the commercial atmosphere is too strong, with many shops lining the route. Prices for food and souvenirs are often double those outside the scenic area. To achieve the strategic goal of making Liyang an "international city of good life resort destination", Tianmu Lake and Nanshan Bamboo Sea, as national 5A-level scenic areas, need to reform and improve their business philosophy and development mode.

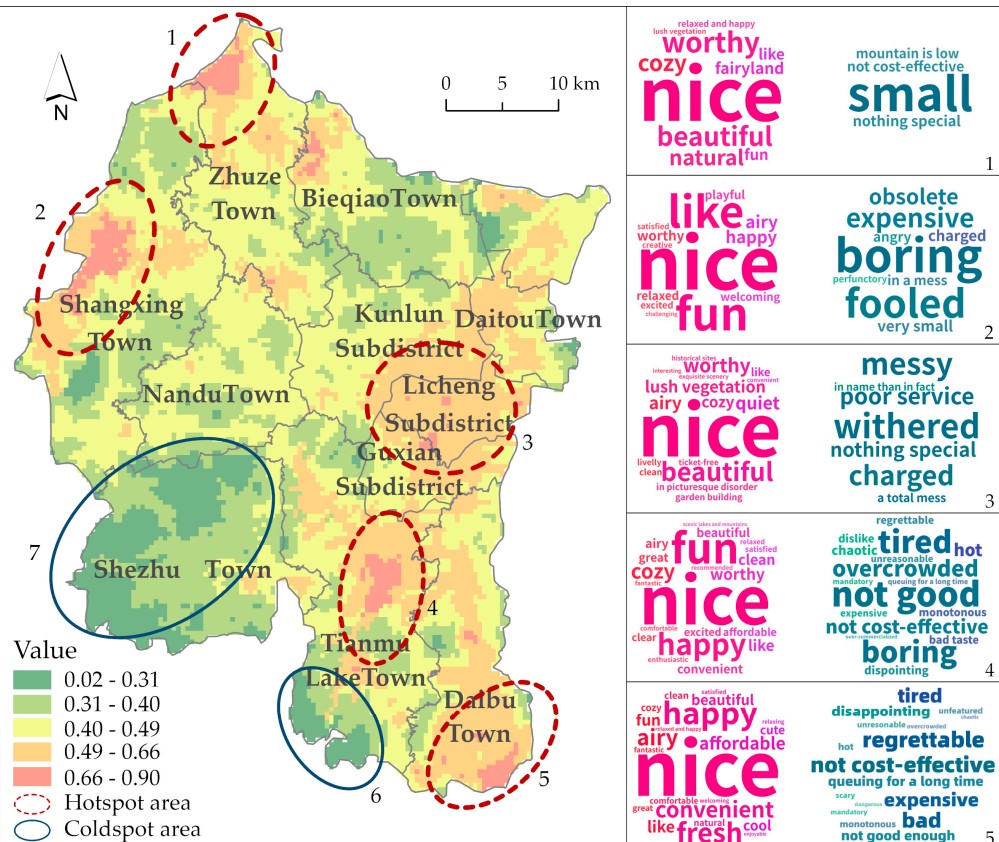

**Figure 11.** Word clouds of hotspot areas. 1 indicates Wawushan Scenic Area, 2 indicates Caoshan Resort, 3 contains the attractions of the central city, 4 mainly covers Tianmu Lake, 5 mainly covers Nanshan Bamboo Sea Scenic Area, 6 indicates a cold spot area in Tianmu Lake Town, and 7 indicates the cold spot area in Shezhu Town.

The area shown as No. 6 in Figure 11 is the cold spot area in Tianmu Lake Town. As shown in Figure 12, the terrain here is hilly and wooded, with a good mountain forest landscape. At the same time, due to the rich mountain resources, there are several mining sites, which not only damage the ecological environment but also affect the scenic quality. Therefore, environmental governance and ecological restoration should be carried out. The waste rock, wastewater, and waste gas generated during the mining process should be treated promptly. Measures such as vegetation restoration and land reclamation should be taken to restore the mountains. In this area, two reservoirs located at the top and the bottom of the mountain make up the Liyang Pumped Storage Power Station, which is a successful example of promoting a low-carbon economy. The opportunity exists to take advantage of this green ecological power station for tourism development. With ecological promotion and technology demonstration as highlights, it could be opened to the public for science education while promoting various activities around the mountain, such as hiking, cycling, and sightseeing.

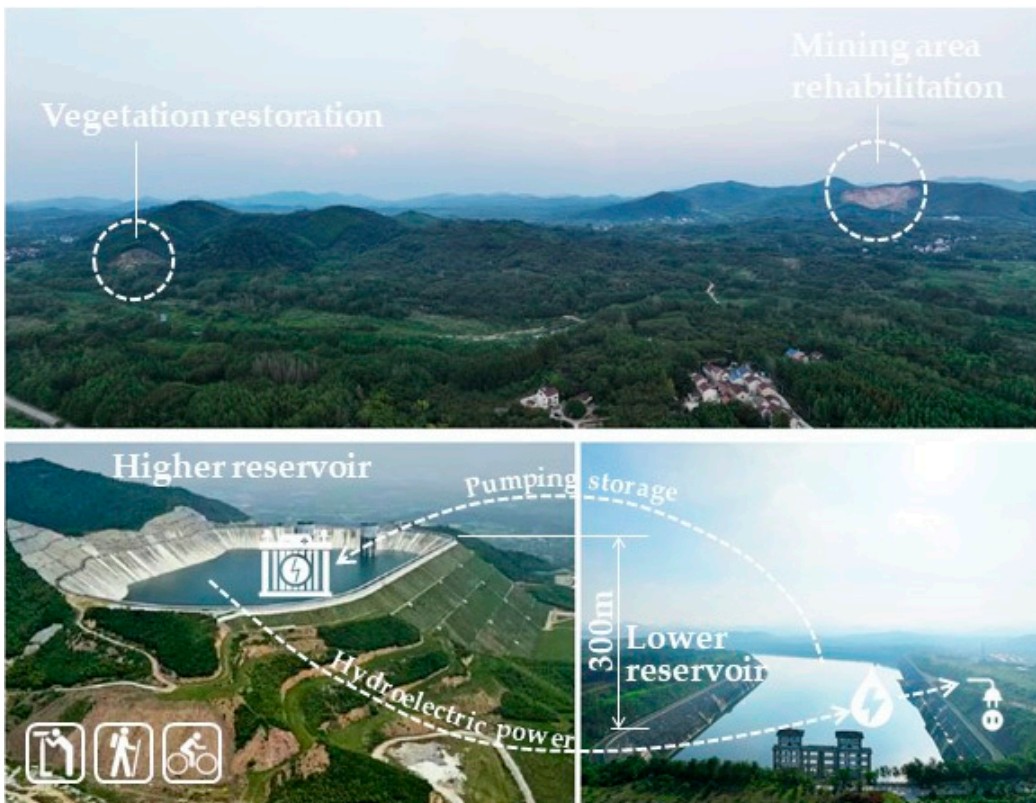

**Figure 12.** Aerial photos and proposed programs for the cold spot area in Tianmu Lake Town.

No. 7 in Figure 11 shows the cold spot area of Shezhu Town. As shown in Figure 13, there is a large area of contiguous aquaculture ponds with beautiful scenery, providing good conditions for the development of agrotourism. For some ponds with poor-quality water, measures such as improving the substrate and strengthening water circulation should be taken for water treatment. In addition to the aquaculture function, the ponds can also be combined with leisure activities to create a tourism growth point that integrates agriculture and tourism, providing a rich experience for visitors. Platforms and walkways can be built on the aquaculture ponds to provide visitors with fishing and shrimping experiences. Larger ponds can be developed into artificial lakes to provide visitors with water entertainment such as canoeing, water skiing, and swimming. Villages around the area can provide support services, including food, drink, rest, and accommodation. Within this cold spot area, there is a cement factory called Jinfeng Cement Group, which covers an area of about 2.5 km² and is the largest cement production enterprise in Jiangsu province. Green production is an important task for industrial upgrading and an effective way to achieve sustainable development. The Jinfeng cement factory should further control pollution and transform the plant into a green factory that can be opened to the public. At the same time, the development history and technical process of the factory can be displayed to the public by building a museum. The proposed programs will make the factory a new tourism attraction combining production, entertainment, and technology.

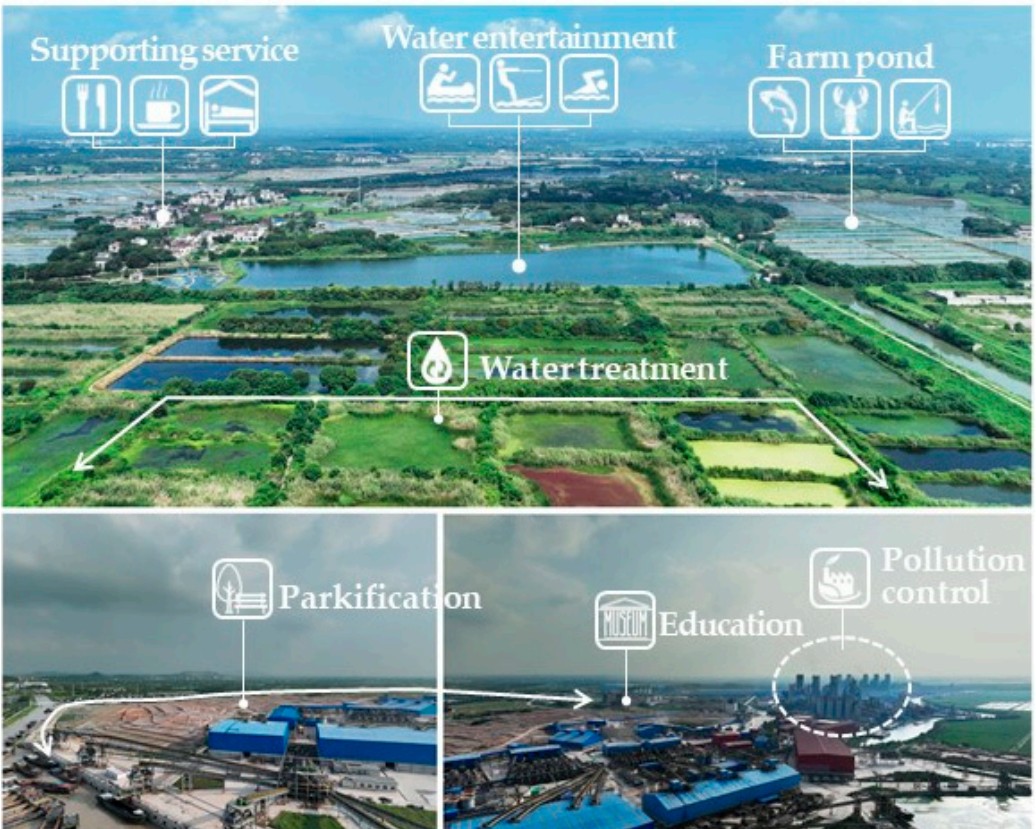

**Figure 13.** Aerial photos and proposed programs for the cold spot area in Shezhu Town.

## 4. Discussion

### 4.1. Practical Implications for Future Tourism Development

In order to promote the coordinated development of the ecological environment and tourism in Liyang, it is necessary to take measures from the perspective of improving ecosystem services. Following a comprehensive evaluation and analysis, the following suggestions are proposed for the city's future tourism development.

(1) For developed scenic spots and attractions, such as the hotspot areas shown in the previous analysis, a large number of online reviews have already been generated. This paper used social media data mining and sentiment analysis to obtain word clouds of tourist sentiment tendencies. The positive word clouds reflect the advantages that should be carried forward. The negative word clouds reflect the dissatisfaction of tourists, which needs to be targeted for improvement and enhancement. First, the tour routes within the scenic areas need to be more reasonable and flexible. Second, the transport capacity within the scenic areas needs to be expanded, especially during the peak season when there are large crowds, to eliminate long waiting times and crowded queues for tourists. In addition, the high fees charged by scenic spots are also a prominent issue, especially for the national 5A-level Tianmu Lake and Nanshan Bamboo Sea. The business model of seeking high profits from ticket sales and transport fees should be changed as soon as possible because tourists will not return. The excellent tourism resources should be fully utilized to develop advanced services and related industry chains, providing tourists with a variety of affordable and high-quality tourism experiences. This will in turn attract a continuous stream of tourists, creating a virtuous cycle and promoting the development of the whole city.

(2) As analyzed in the previous section, the cold spots of tourism service are located in the southwest of the city, where there are good mountain and forest landscapes, as well as large areas of farmland and ponds. It is, therefore, necessary to carry out mountain restoration and farmland system restoration in these areas. At the same time, tourism

should be combined with local industries such as the pumped-storage power station, the cement factory, and shrimp farming. The opportunity exists to develop new tourism projects with a beautiful ecological environment and unique characteristics.

(3) At present, Liyang has more than 20 villages with national or provincial honors. However, these well-developed villages are small in scale and scattered in distribution, failing to form a clustering effect. In the future, it will be necessary to make greater efforts to build rural tourism destinations that are highly integrated with ecology, production, and life. It is also important to create a rural tour route that connects the villages, which will become an important aspect of all-for-one tourism.

### 4.2. Methodological Advantages

As there has been controversy over whether tourism service belongs to cultural services, few articles have studied tourism from the perspective of ecosystem services. Through reviewing the literature and authoritative reports, this paper argues that tourism service should be treated as a cultural service for tourism-oriented cities. Taking Liyang, a typical all-for-one tourism city in China, as an example, a tourism service evaluation indicator system was established from three aspects: the quality of tourism resources, the comprehensiveness of tourism service facilities, and tourist satisfaction. This study expands the boundary of cultural services and enriches the research cases of ecosystem service evaluation.

In terms of analyzing tourist satisfaction, this paper used the Octopus tool to collect big data on tourist check-in photos and comments from tourism websites and used ROST CM for text mining to identify tourists' sentiment tendencies. When analyzing the hotspot areas, the big data for tourism reviews is segmented and processed to generate positive and negative word clouds, based on which the corresponding improvement suggestions were proposed. Instead of traditional questionnaires, the big data analysis method can collect multi-source data of larger volume, which is not only cost-effective but also avoids the subjective bias caused by a small sample size.

### 4.3. Limitations and Future Research Directions

Given the data availability, this study selected seven indicators to develop the basic evaluation system; however, other indicators should be further considered to improve the research framework. For example, the landscape aesthetic quality should also be included in the scope of tourism resource quality. Each aspect of tourism service, such as dining, accommodation, recreation, and medical services, should be specified with indicators. In addition, when measuring tourism accessibility, not only should objective accessibility be considered, but also perceived accessibility, which emphasizes the ease of access to activities, should be considered [66].

Although big data analysis is more objective than questionnaires, there is still a sample representation issue. In general, social media users tend to be younger, more educated, and urban residents [67,68], which may lead to an overestimation of the perceptions of certain groups. In contrast, questionnaires and interviews have the advantage of being able to collect demographic information. Future research should organically combine conventional and innovative social survey methods for spatial and non-spatial data collection.

### 5. Conclusions

With many advantages, such as good economic benefits, low resource consumption, and low environmental impact, tourism is an important industry of the national economy and an important channel for expanding employment [69]. As a cross-regional activity, tourism plays a crucial role in promoting economic development, human well-being, and cultural awareness. At present, there is still no unified understanding of tourism service in ESs research, and no evaluation method system for tourism service has been formed. This paper advocates that tourism service should be considered as a cultural service, at least for tourism cities. Liyang in China was taken as an example to evaluate tourism service using

technical methods such as GIS spatial analysis, social media data mining, and sentiment analysis. The research results show the following conclusions:

(1) The areas with higher tourism service are distributed in the northwestern edge and the east of the city, while the areas with lower tourism service are mainly distributed in the northeastern and southwestern parts of the city. The hotspot areas are located in the Wawushan Scenic Area and Bieqiao Village in the northern part of the city, Caoshan Resort in the west, Tianmu Lake and Nanshan Bamboo Sea in the south, and the central area of the city. The cold spot areas are concentrated in the southwest.

(2) The hotspot areas should focus on improving the tour routes, transport capacity, and excess charges. The cold spot areas should work on ecological restoration and, meanwhile, create new tourism attractions by combining the local industries. Moreover, rural tourism needs further development in the future.

(3) For tourism-oriented cities, tourism service should be considered as the dominant type of cultural service, and cities should be assessed from an ESs perspective in order to better achieve harmonious development of both the ecosystem and the tourism industry.

**Author Contributions:** Conceptualization, X.F. and Y.C.; methodology, validation, data curation, writing—original draft preparation, X.F.; project administration, Y.C.; software, X.F.; and writing—review and editing, supervision, X.F. All authors have read and agreed to the published version of the manuscript.

**Funding:** This research received no external funding.

**Data Availability Statement:** Not applicable.

**Acknowledgments:** Ethical considerations were taken into account throughout this research. Personal information from social media platforms was anonymized to ensure confidentiality and direct quotes were used with caution to avoid compromising the identity of informants. In addition, we would like to thank the reviewers for providing constructive comments on this manuscript.

**Conflicts of Interest:** The authors declare no conflict of interest.

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
