# Peer review of "Assessing a Tourism City from an Ecosystem Services Perspective: The Evaluation of Tourism Service in Liyang, China"

_land, doi:10.3390/land12112019_

Round 1

Reviewer 1 Report

Comments and Suggestions for Authors

The authors have used a combination of GIS spatial analysis and big data text mining to evaluate the tourism service from three aspects: the quality of tourism resources, the comprehensiveness of tourism service facilities, and the satisfaction of tourists. However, the aim of the research is not defined neither in the abstract, nor in the introduction.  There is no hypothesis or research questions defined in the research, therefore the paper lacks clear focus and a reader does not understand, what is the main aim/objective of the research. The authors use multiple research methods, and refer to many international and relevant studies.

The title should not comprise an abbreviation (ES in this case), so I recommend to change ES in the title to full words "ecosystem services". All figures and tables are relevant, useful and accurate. Information about the research ethics must be included.

Conclusions must comprise some discussion part: the authors should point to the comparison of their own results with the literature review. Also after defining the aim or the research and hypothesis, conclusions reflect on whether results support hypothesis or not.

Author Response

Thanks very much for  the helpful comments and suggestios. We have followed the suggestions to make the overall revision of the paper. Please see the attatchment.

Reviewer 2 Report

Comments and Suggestions for Authors

This paper presents research results on the evaluation of tourism services in Liyang, China, using technical methods such as GIS spatial analysis, social media data mining, and sentiment analysis. It should be emphasized that the topic is very interesting, important and current. The multi-directionality (many sources of information) of the analyzes conducted is noteworthy.

The research results are presented in a detailed but clear and accessible way.

A valuable element of the article are also the practical implications for the future development of tourism in the studied region.

It is also pleasing that the authors are aware of the limitations of their research.

A few minor shortcomings (which do not have a major impact on the high quality and value of this article) are indicated in the comments in the text.

Author Response

Thanks very much for the helpful comments and suggestions. We have followed the suggestions to make the overall revision of the paper. Please see the attachment.
